# Evaluation of the Effect of Storage Methods on Fecal, Saliva, and Skin Microbiome Composition

Clarisse Marotz,[a] Kellen J. Cavagnero,[a,d] Se Jin Song,[b] Daniel McDonald,[a] Stephen Wandro,[b] Greg Humphrey,[a] MacKenzie Bryant,[a] Gail Ackermann,[a] Edgar Diaz,[a] Rob Knight[a,b,c]

[a]Department of Pediatrics, University of California San Diego, La Jolla, California, USA
[b]Center for Microbiome Innovation, University of California San Diego, La Jolla, California, USA
[c]Department of Computer Science and Engineering, University of California San Diego, La Jolla, California, USA
[d]Department of Dermatology, University of California San Diego, La Jolla, California, USA

Clarisse Marotz and Kellen J. Cavagnero contributed equally to this work. Author order was determined by submission organization and formatting time.

**ABSTRACT** As the number of human microbiome studies expand, it is increasingly important to identify cost-effective, practical preservatives that allow for room temperature sample storage. Here, we reanalyzed 16S rRNA gene amplicon sequencing data from a large sample storage study published in 2016 and performed shotgun metagenomic sequencing on remnant DNA from this experiment. Both results support the initial findings that 95% ethanol, a nontoxic, cost-effective preservative, is effective at preserving samples at room temperature for weeks. We expanded on this analysis by collecting a new set of fecal, saliva, and skin samples to determine the optimal ratio of 95% ethanol to sample. We identified optimal collection protocols for fecal samples (storing a fecal swab in 95% ethanol) and saliva samples (storing unstimulated saliva in 95% ethanol at a ratio of 1:2). Storing skin swabs in 95% ethanol reduced microbial biomass and disrupted community composition, highlighting the difficulties of low biomass sample preservation. The results from this study identify practical solutions for large-scale analyses of fecal and oral microbial communities.

**IMPORTANCE** Expanding our knowledge of microbial communities across diverse environments includes collecting samples in places far from the laboratory. Identifying cost-effective preservatives that will enable room temperature storage of microbial communities for sequencing analysis is crucial to enabling microbiome analyses across diverse populations. Here, we validate findings that 95% ethanol efficiently preserves microbial composition at room temperature for weeks. We also identified the optimal ratio of 95% ethanol to sample for stool and saliva to preserve both microbial load and composition. These results provide rationale for an accessible, nontoxic, cost-effective solution that will enable crowdsourcing microbiome studies, such as The Microsetta Initiative, and lower the barrier for collecting diverse samples.

**KEYWORDS** 16S rRNA gene amplicon sequencing, benchmarking, metagenomics, microbiome, preservation

Next generation sequencing has enabled an unprecedented view of human-associated microbial communities. This insight has been leveraged to improve understanding of how such communities contribute to human health. Many investigations focus on the gut, using fecal samples as a proxy for the intestinal microbiome, but skin and oral samples are also commonly collected for human microbiome analysis. To comprehensively understand human-associated microbiomes, samples have been acquired from healthy and diseased patients in the clinical setting (1), indigenous peoples in the field (2), and individuals across the globe through community science

Address correspondence to Rob Knight, robknight@ucsd.edu.

Simple 95% EtOH preserves fecal and saliva samples for microbiome analysis for at least 8 weeks at room temperature.

efforts (3–5). In the clinic, samples are typically frozen immediately at −20°C or below until nucleic acid extraction and sequencing, the gold standard in the microbiome field. However, beyond the clinic, immediate freezing is often not possible, thus leading to the use of preservatives. Importantly, there is currently no standard protocol for microbiome preservation. Identifying a cost-effective, practical solution to microbiome sample preservation could help reduce the barrier to collecting specimens from geographic locations that contain high microbial diversity but are vastly underrepresented (6).

Numerous studies have aimed to understand how fecal storage affects microbiome composition (7–10). In 2016, our group reported one of the largest investigations thus far (11). We performed 16S rRNA gene amplicon sequencing on over 1,200 samples, which included fecal samples from 15 individuals subjected to six preservatives, six storage temperatures, and four time points. Our analysis revealed that composition was well maintained over time and across temperatures when samples were stored in 95% ethanol.

In the last 4 years, microbiome sequencing techniques and the computational tools to analyze next generation sequencing data have advanced considerably. Here, we aimed to improve the understanding of how storage affects microbiome composition using state-of-the-art methods. First, we reanalyzed our original 16S rRNA gene amplicon sequencing data set using a high-resolution exact sequence variant method (12, 13). Next, to further increase resolution (14), we performed shallow shotgun metagenomic sequencing on archived DNA from our original study. Results from these studies confirmed our original conclusion that 95% ethanol is efficacious in preserving fecal microbiome integrity. Finally, we extended our study of 95% ethanol to investigate optimal sample collection protocols for fecal, saliva, and skin samples.

## RESULTS

**95% EtOH is an efficient room temperature preservative for microbiome analyses.** We first set out to determine the effect of sample storage on fecal microbial composition. To that end, we reanalyzed 16S rRNA gene amplicon sequencing (16S) data from our previous storage study (11). Because shotgun metagenomic sequencing provides deeper characterization of microbial communities and does not necessarily reproduce results from 16S analyses (14), we also performed shotgun metagenomic sequencing on remaining nucleic acid extracts. In the prior study, fecal samples had been obtained from 15 subjects (10 humans and 5 dogs) and subjected to six different preservative conditions (no preservative [None], 70% ethanol [EtOH], 95% ethanol, RNAlater, OMNIgene GUT [OMNI], and FTA cards [FTA]), up to six different temperature conditions (ambient, −20°C, −20°C after 1 week, 4°C, freeze-thaw, and heat), and up to four time points (fresh, 1 week, 4 weeks, and 8 weeks).

We assessed phylogenetically informed beta-diversity in our shotgun metagenomic sequencing data set using a curated microbial phylogenetic tree (15) (see Materials and Methods). In both 16S (see Fig. S1A in the supplemental material) and shotgun data sets (Fig. 1A), weighted UniFrac beta-diversity was driven primarily by subject and subject species, rather than by whether samples were stored or not, or by specific storage method. This finding held true across six different beta-diversity metrics (see Table S1 in the supplemental material).

To determine which storage method induced the least amount of change in fecal microbiome composition, we compared weighted UniFrac distances between each subject's fresh and stored samples. As expected, immediate sample freezing (–20°C) resulted in the least change across conditions with both shotgun (Fig. 1B) and reprocessed 16S data (Fig. S1B). As in our original 16S analysis (11), samples left unfixed or preserved in 70% ethanol exhibited the greatest compositional change with storage, while 95% ethanol, FTA, OMNI-gene GUT, and RNAlater minimized changes in composition.

To identify potential changes in the relative abundance of specific taxonomic clades, we compared the relative abundance of each genus in each subject's sample processed fresh versus 8 weeks out for each preservative (Fig. 1C, Fig. S1C). As expected, the correlation between fresh and 8-week samples was markedly reduced

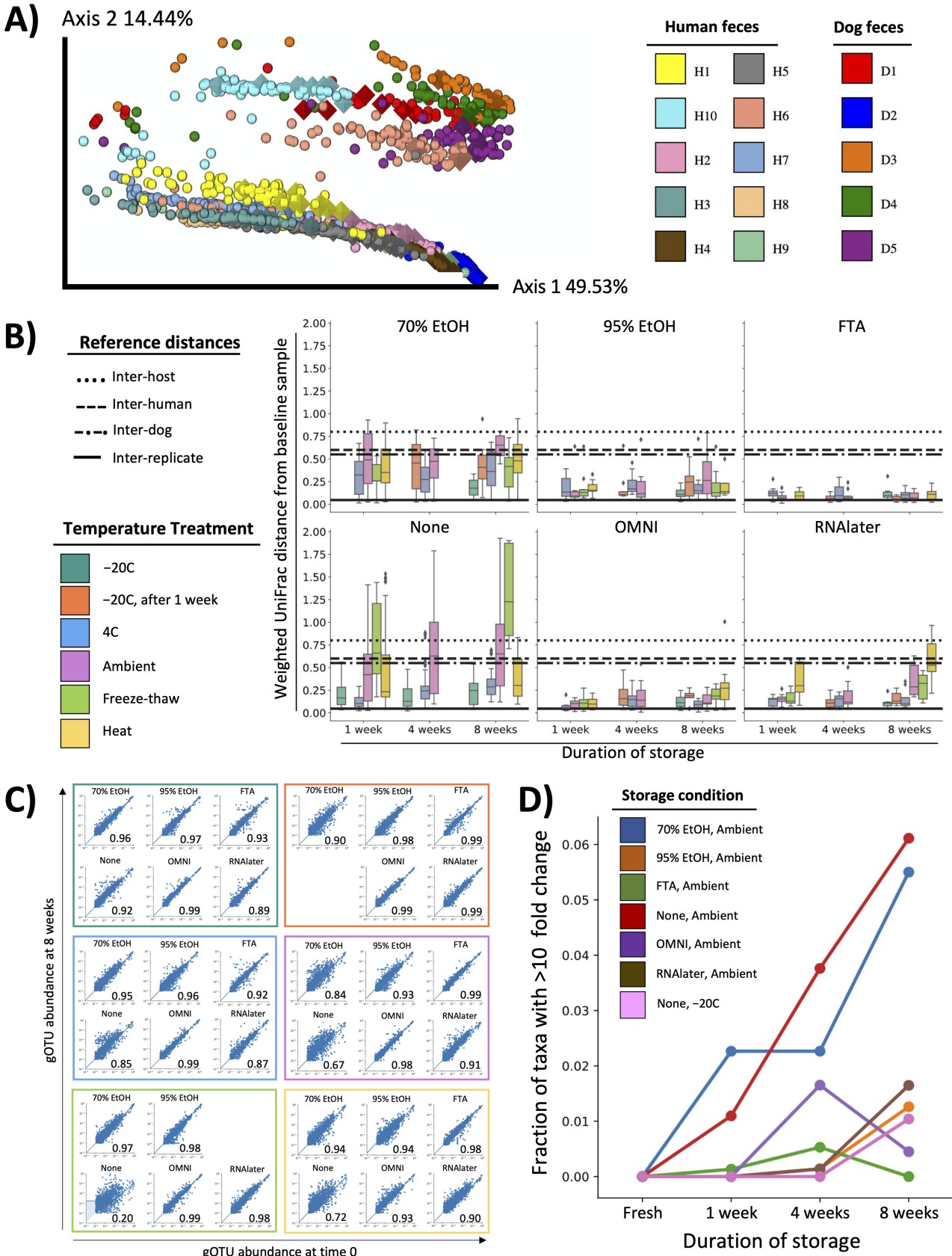

**FIG 1** Effect of storage on fecal microbiome composition with shallow shotgun metagenomic sequencing. (A) Beta-diversity of shotgun metagenomic sequencing data estimated with weighted UniFrac, colored by participants. Diamonds represent fresh samples. (B) Weighted UniFrac

mSystems®

without the use of a fixative—unless the sample was immediately frozen (−20°C). Samples stored in 95% ethanol, FTA, and OMNI-gene GUT had similarly tight taxonomic correlations across conditions, while 70% ethanol-treated samples correlated relatively poorly under several conditions.

Previous reports have shown microbial blooms occur over time in fecal samples without the use of preservatives (11, 16). Therefore, we determined the percentage of taxa with a greater than 10-fold change in relative abundance over time for each stored sample. As expected, microbial blooms occurred in samples unfixed and stored ambiently, and blooming was prevented by immediately freezing (Fig. 1D, Fig. S1D). As in our previous report, we found that samples stored in 70% ethanol contained similar levels of blooming organisms to unfixed samples, whereas samples stored in 95% ethanol, OMNI-gene GUT, FTA, and RNAlater contained levels of blooming organisms similar to those of immediately frozen samples.

As with composition, differences in Shannon and Simpson alpha-diversity across storage times were largest when samples were unfixed or stored in 70% EtOH, and smallest when stored in 95% EtOH, OMNI-gene GUT, FTA, and RNAlater (Fig. S2). To determine the ability to correctly classify samples after preservation, we trained a random forest sample classifier on subjects' fresh samples (none, fresh, and ambient) and determined the accuracy of the classifier in predicting the subject from which each stored sample was derived for each storage condition. The classifier was nearly perfect at predicting sample donors across storage methods (Fig. S3). Of the few misclassified samples, most were not immediately frozen or fixed in 95% ethanol.

Collectively, our findings indicate that 95% ethanol, OMNI-gene GUT, FTA, and RNAlater are effective fecal microbiome preservatives. In our view, 95% ethanol is the most practical as it is nontoxic and cost-effective. Because our reanalyzed 16S data and shotgun metagenomic data were highly consistent (Fig. S4, Mantel Spearman correlation = 0.75), we performed a follow-up experiment with 16S rRNA gene amplicon sequencing to determine the optimal sample to 95% ethanol ratio for sample collection.

**Optimizing the ratio of 95% ethanol preservative to sample.** To determine the optimal ratio of 95% EtOH to fecal sample for microbiome analyses, fecal samples from 12 participants were divided into seven categories for comparison (Fig. 2A); a fecal sample was swabbed following the American Gut Project protocol (4) and immediately frozen, a 1-g aliquot of the fecal sample was frozen immediately with no preservative, a 1-g aliquot of fecal sample was stored in 1 ml, 2 ml, or 5 ml of 95% ethanol and left at room temperature, or a fecal swab was stored in 1 ml of 95% ethanol and left at room temperature, and sample was extracted from the swab or remaining 95% ethanol eluent.

Microbial load was evaluated via quantitative PCR (qPCR) of the 16S rRNA gene. Fecal swabs stored in 95% ethanol and extracted from the swab head had the closest amount of microbial load to the gold standard out of all the samples stored at room temperature (Fig. 2B). Principal coordinate analysis (PCoA) calculated with weighted UniFrac revealed that samples tended to cluster by host subject rather than storage method (Fig. 2C). Permutational multivariate analysis of variance (PERMANOVA) analysis across all distance metrics tested confirmed that beta-diversity was more strongly driven by host subject rather than storage method (Table S2). The relative abundance of each genus was compared between the gold standard (swab immediately frozen) and each storage method. Relative abundances of all storage methods were significantly correlated with the gold standard (Fig. 2D), and swab stored in 95% ethanol had one of the strongest correlations (Pearson $R = 0.973$).

**FIG 1** Legend (Continued)
distances between stored samples and baseline samples (fresh, ambient) for each preservative. (C) Pearson correlation where each point represents a genus with the relative abundance in the fresh fecal samples on the $x$ axis and the relative abundance in the stored fecal sample stored for 8 weeks on the $y$ axis, with each plot series representing a different storage temperature. Both axes are presented in log base 10 scale. (D) Fraction of taxa with greater than 10-fold change relative to fresh samples for each storage condition.

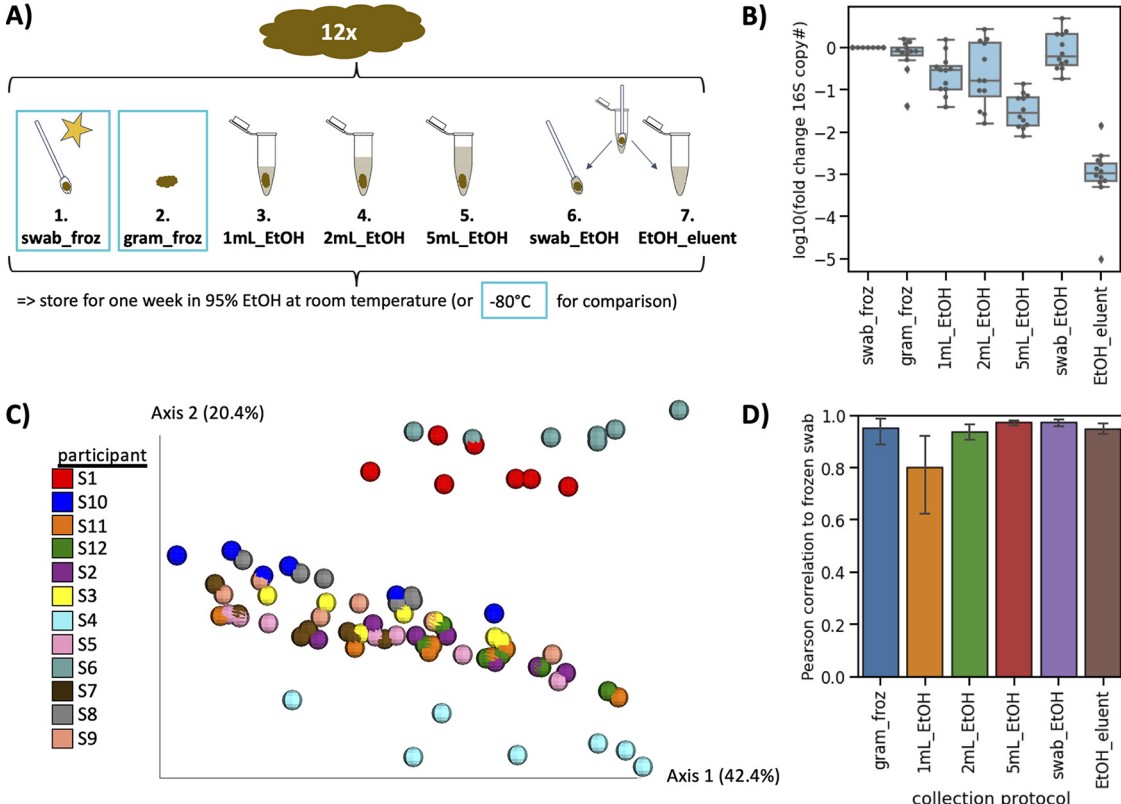

**FIG 2** Identifying optimal collection protocol for fecal samples in 95% ethanol. (A) Experimental design. Fecal samples were collected from 12 participants, and each was divided into seven groups; the fecal sample was swabbed (step 1) or 1 g aliquoted (step 2) and immediately frozen in −80°C. The remaining samples were stored at different ratios with 95% ethanol and kept at room temperature for 1 week; 1 g feces in either 1 ml (step 3), 2 ml (step 4), or 5 ml (step 5) of 95% ethanol, or fecal sample swabbed and stored in 1 ml 95% ethanol. After 1 week, DNA was extracted from each sample (and for the swab in ethanol, the swab [step 6] and remaining ethanol [step 7] were independently extracted). (B) qPCR on the extracted DNA using primers against the 16S rRNA gene normalized to the frozen swab. (C) Beta-diversity calculated with unweighted UniFrac and colored by participant. (D) Pearson correlation of genus relative abundance in each sample compared to the frozen swab. Error bars represent standard errors of the means across the 12 participants.

Next, we applied the same experimental design to oral microbiome samples. Twelve participants provided unstimulated saliva samples which were stored in seven different ways for comparison (Fig. 3A). A saliva sample was swabbed following the American Gut Project (AGP) protocol (4) and immediately frozen, a 500-$\mu$l aliquot of the saliva sample was frozen immediately with no preservative, a 500-$\mu$l aliquot of saliva sample was stored in 0.5 ml, 1 ml, or 2 ml of 95% ethanol and left at room temperature, or a saliva swab was stored in 1 ml of 95% ethanol and left at room temperature, and a sample was extracted from the swab or remaining 95% ethanol eluent.

qPCR analysis revealed that saliva samples stored in 95% ethanol at a ratio of 1:1 or 2:1 had the closest amount of microbial load to the gold standard out of all the samples stored at room temperature (Fig. 3B). Saliva swabs stored in 95% ethanol had significantly lower 16S rRNA gene copies whether the sample was extracted from the swab head or the remnant 95% ethanol. Similar to fecal samples, PCoA of the saliva samples revealed that beta-diversity was more strongly driven by host subject rather than storage method across all distance metrics (Fig. 3C, Table S2). The genus-level relative abundances of all storage methods were significantly correlated with the gold standard (Fig. 3D), and saliva samples were stored in 95% ethanol at a ratio of 1:1 (Pearson $R = 0.88$), 2:1 (Pearson $R = 0.89$), and 4:1 (Pearson $R = 0.91$) had the strongest correlations.

Finally, we applied the same experimental design to skin microbiome samples. Twelve participants provided triplicate skin swabs each from their forehead and right palm, which were then stored in three different ways for comparison (Fig. 4A). Skin swabs were either

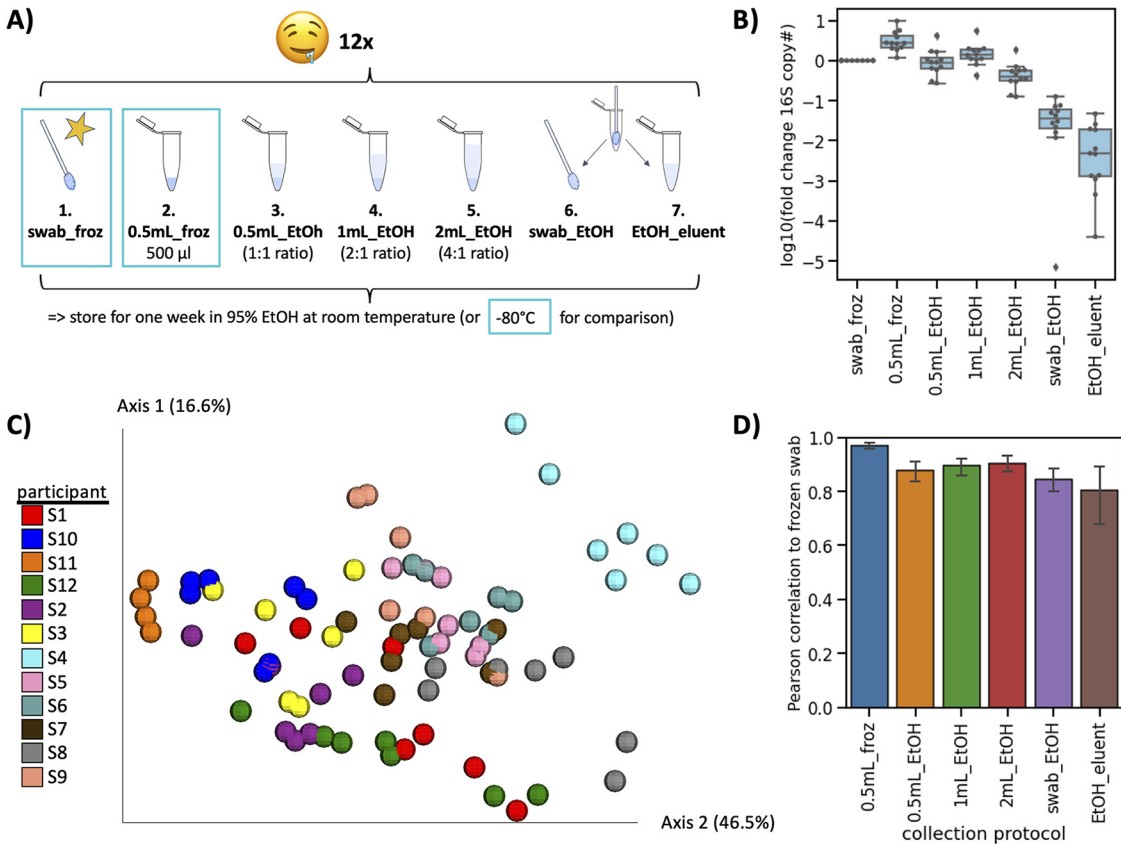

**FIG 3** Identifying optimal collection protocol for saliva samples in 95% ethanol. (A) Experimental design. Unstimulated saliva samples were collected from 12 participants, and each was divided into seven groups. The saliva sample was swabbed (step 1) or 500 μl aliquoted (step 2) and immediately frozen in −80°C. The remaining samples were stored at different ratios with 95% ethanol and kept at room temperature for 1 week: 500 μl saliva in either 0.5 ml (step 3), 1 ml (step 4), or 2 ml (step 5) of 95% ethanol, or saliva sample was swabbed and stored in 1 ml of 95% ethanol. After 1 week, DNA was extracted from each sample (and for the swab in ethanol, the swab [step 6] and remaining ethanol [step 7] were independently extracted). (B) qPCR on the extracted DNA using primers against the 16S rRNA gene normalized to the frozen swab. (C) Beta-diversity calculated with weighted UniFrac and colored by participant. (D) Pearson correlation of genus relative abundance in each sample compared to the frozen swab. Error bars represent standard errors of the means across the 12 participants.

immediately frozen or stored in 1 ml 95% ethanol and left at room temperature, and sample was extracted from the swab or remaining 95% ethanol eluent.

qPCR analysis revealed that both forehead and palm swab samples stored in 95% ethanol had significantly reduced microbial loads compared to swabs stored at −80°C without preservative (Fig. 4B). Similar to saliva and fecal samples, beta-diversity was most strongly driven by storage method across all distance metrics calculated (Fig. 4C, Table S2). The genus-level relative abundances were correlated better with the gold standard in forehead samples extracted from swab (Pearson $R = 0.94$) than ethanol (Pearson $R = 0.70$). This held true for palm samples (swab extraction Pearson $R = 0.78$, ethanol extraction Pearson $R = 0.61$), although these correlated less well than forehead samples (Fig. 4D).

## DISCUSSION

Efficient, cost-effective preservatives for microbiome analyses are increasingly important as the scientific community expands sampling efforts to include more diverse communities and organisms. Here, we used a state-of-the-art denoising tool to reanalyze amplicon sequencing data from a large comparative study (11). We also generated shotgun metagenomic sequencing on these remnant samples. Both the updated 16S sequencing results and the shotgun metagenomics validated our previous finding that 95% ethanol performs well to preserve microbiome samples when immediate, consistent freezing is not an option.

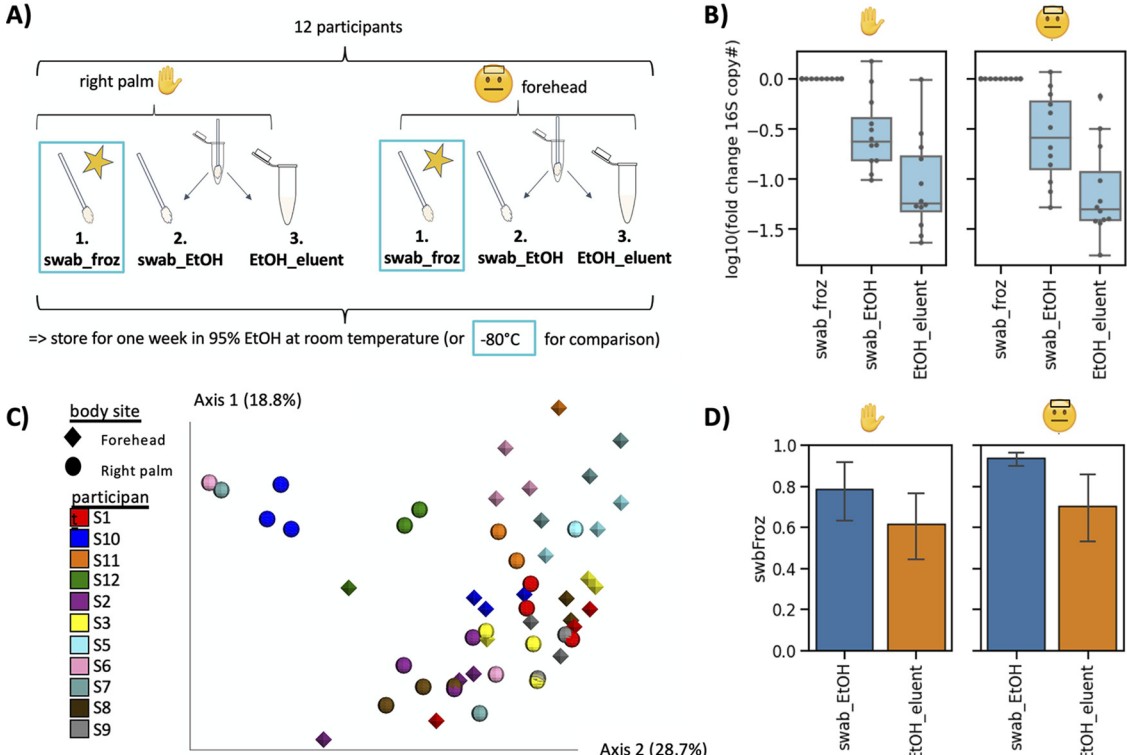

**FIG 4** Identifying optimal collection protocol for skin samples in 95% ethanol. (A) Experimental design. Forehead and right palm skin swabs were collected from 12 participants, and each was divided into three groups; one swab was immediately frozen in −80°C (step 1). The second swab was stored in 95% ethanol and kept at room temperature. After 1 week, DNA was extracted from the frozen swab, the swab in ethanol the swab (step 2) and remaining ethanol (step 3). (B) qPCR on the extracted DNA using primers against the 16S rRNA gene normalized to the frozen swab for both the palm (left) and forehead (right) samples. (C) Beta-diversity calculated with unweighted UniFrac and colored by participants. Diamonds represent forehead swabs, and spheres represent palm swabs (D) Pearson correlation of genus relative abundance in each sample compared to the frozen swab. Error bars represent standard errors of the means across the 12 participants.

We performed follow-up experiments to determine the optimal ratio of sample to 95% EtOH for different sample types, including fecal, saliva, and skin samples. For fecal samples, storing a swab of feces in 95% EtOH and extracting from the fecal swab provided the best recovery of biomass and high fidelity of microbial composition compared to the gold standard of freezing. For saliva samples, the best solution was storing liquid saliva at a ratio of 1:2 with 95% EtOH. Skin sample preservation did not work nearly as well at retaining biomass or microbial composition as saliva and fecal samples. One reason could be that the primers targeting the V4 region of the 16S rRNA gene used in this study are suboptimal for skin amplicon sequencing (17); therefore, these results may need to be validated via shotgun metagenomic sequencing. However, this is complicated because skin samples are extremely low biomass and contain a high percentage of host DNA (18), underscoring the difficulty of designing a standardized skin sample collection method for high-throughput analysis.

Together, our findings support the use of 95% EtOH for fecal and saliva sample preservation for downstream microbiome sequencing analyses. These results will guide collection protocols and improve our ability to collect samples for microbiome analysis when immediate freezing is not available.

## MATERIALS AND METHODS

**Validation of Song et al. study with updated techniques.**

**(i) Shotgun metagenomic sequencing.** Remnant genomic DNA (gDNA) from Song et al. (11) was used for shotgun metagenomic sequencing as previously described (19). Briefly, gDNA was quantified with Quant-iT PicoGreen double-stranded DNA (dsDNA) assay kit (ThermoFisher Scientific), and 1 ng of

input DNA was used in a 1:10 miniaturized Kapa HyperPlus protocol. For samples with less than 1 ng DNA, a maximum volume of 3.5 $\mu$l input was used. Equimolar amounts of each sample were pooled, and the library was size selected for fragments between 300 and 700 bp on the Sage Science PippinHT. The pooled library was sequenced as a paired-end 150-cycle run on an Illumina HiSeq2500 v2 run at the University of California San Diego (UCSD) IGM Genomics Center.

**(ii) Shotgun metagenomic and 16S analysis.** Shallow shotgun metagenomic reads were aligned using SHOGUN (20) with the default settings against the Web of Life database (https://github.com/qiyunzhu/woltka) and clustered using the genome operational taxonomic unit (gOTU) pipeline (https://github.com/qiyunzhu/woltka). For 16S rRNA gene amplicon sequencing (16S), reads were demultiplexed, quality control filtered, and trimmed to 100 nucleotides (nt) using Qiita (study identifier [ID] 10934). Reads were then denoised into amplicon sequence variants using Deblur v1.1.0 (13). Fragments were inserted into the Greengenes 13.8 reference phylogeny (21) using SEPP (22) with QIIME 2's (23) q2-fragment-insertion plugin (24). QIIME 2 version 2019.7 was used throughout our validation of the Song et al. study.

**(iii) Diversity measures.** 16S and shotgun metagenomic data were first rarefied to 10,000 and 250,000 reads, respectively. Alpha-diversity (Shannon, Simpson) and beta-diversity (unweighted UniFrac [25], weighted UniFrac, Bray Curtis, Jaccard, Aitchison [26], robust principal component analysis [RPCA] [27]) were calculated using the QIIME 2's q2-diversity plugins "alpha-group-significance" and "beta-group-significance," respectively. EMPeror (28) plots were visualized using QIIME2 view. To understand the effect of each preservative on bacterial composition and diversity, we plotted the weighted UniFrac distance and Shannon diversity difference between individuals' preserved samples and nonpreserved, fresh samples.

**(iv) Relative abundance analyses.** To determine sample relative abundance, reads were first classified by taxon using the QIIME 2 plugin q2-feature-classifier (29) with the "classify-sklearn" action against the default Greengenes 13.8 classifier, and then collapsed to genus level using the QIIME 2 plugin q2-taxa with the "collapse" action. For each preservative and storage temperature, genus level relative abundances from fresh samples were plotted against 8-week-old samples, and Pearson correlation coefficients were calculated. To determine the impact of each preservative on microbial bloom formation over time, the fraction of taxa with greater than 10-fold change relative to fresh samples was calculated for each preservative (at ambient temperature) at each time point. As a control, the fraction of taxa with greater than 10-fold change relative to fresh samples was calculated for frozen samples at each time point.

**(v) Sample classification.** A supervised learning approach was used to determine whether stored samples were similar enough to fresh samples to predict the donor of each stored sample. We trained a model on fresh replicate samples (six per individual) and determined the model's ability to accurately classify individuals for each storage condition. To that end, we utilized the QIIME 2's q2-sample-classifier (30) plugin with the "classify-samples" action.

**(vi) Procrustes analysis.** To directly compare the beta-diversity of the 16S and shotgun data sets, we used the QIIME 2 q2-diversity plugin with the "procrustes-analysis" action using unweighted UniFrac data.

**95% ethanol ratio optimization experiments.**

**(i) Sample collection and storage.** We recruited 12 participants under institutional review board (IRB) 150275 to provide stool, saliva, and skin samples for the sample collection protocol optimization portion of this study. For stool samples, full fecal specimens were collected into commode specimen collection tubs (Fisher Scientific, catalog number 02-544-208) and were processed for storage within 1 h of collection; one gram of feces stored at −80°C, one fecal swab (BD Sterile Falcon Swube, catalog number 220090) stored at −80°C, one gram of feces stored in either 1 ml, 2 ml, or 5 ml of 95% EtOH at room temperature, and one fecal swab in 1 ml of 95% EtOH at room temperature. For saliva samples, participants were asked to provide 4 ml of saliva into a sterile 15-ml conical tube. All saliva samples were vortexed well and processed for storage within 1 h of collection; 0.5 ml stored at –80°C, saliva swab (BD Sterile Falcon Swube, catalog number 220090) stored at –80°C, 0.5 ml saliva stored in either 0.5 ml, 1 ml, or 2 ml 95% EtOH, and one saliva swab stored in 1 ml of 95% EtOH at room temperature. For skin samples, participants were asked to rub a dry, double-headed swab (BD Sterile Falcon Swube, catalog number 220090) against their right palm for 30 s. One swab head was immediately frozen at –80°C, and the other was stored in 1 ml of 95% EtOH. This process was repeated on the forehead. All samples were extracted 7 or 8 days after sample collection as described below.

**(ii) gDNA extraction and 16S rRNA gene amplicon sequencing.** Samples were extracted using the Qiagen PowerSoil MagAttract DNA kit as previously described (31). For swab samples, the entire swab head was broken off into the extraction vial. For the 95% EtOH eluent samples, 400 $\mu$l was added to the extraction vial. For all other samples (fecal or saliva aliquots in 95% EtOH), the sample was swabbed and the head broken off into the extraction vial. 16S rRNA gene amplification was performed according to the Earth Microbiome Project protocol (32). Briefly, Illumina primers with unique reverse primer barcodes (33) were used to amplify the V4 region of the 16S rRNA gene (515fbc-806r [34]). Amplification was performed in a miniaturized volume (35), with single reactions per sample (36). Equal volumes of each amplicon were pooled, and the library was sequenced on the Illumina MiSeq sequencing platform with a MiSeq reagent kit v2 and paired-end 150-bp cycles. Raw data and associated feature tables are publicly available in Qiita (qiita.ucsd.edu) as study ID 12610.

**(iii) Microbial load assessment via qPCR.** To estimate microbial load, sample gDNA was evaluated in triplicate with KAPA Universal qPCR Master Mix (catalog number KK4828) using the Bakt 341F-805R primers (Bakt_341f [5′-CCTACGGGNGGCWGCAG-3′] and Bakt_805R [5′-GACTACHVGGGTATCTAATCC-3′])

(37). Amplification was performed in triplicate 10-$\mu$l reaction mixtures each containing 5 $\mu$l KAPA MasterMix, 0.5 $\mu$l primer mix containing 5 $\mu$M forward and reverse primers, 2 $\mu$l gDNA, and 2.5 $\mu$l H$_2$O. The PCR mix was cycled through the following temperatures on a Bio-Rad CFX real-time PCR system: (i) 95 for 5 min; (ii) 40 cycles with 1 cycle consisting of 95°C for 30 s and 60°C for 30 s; and (iii) 4°C hold. The median value of the triplicate reactions was normalized to the value for the gold standard (frozen swab) for each sample.

**Data availability.** Data are publicly available in Qiita (38) under study IDs 10394 (validation of Song et al.) and 12610 (95% EtOH sample collection study) and through the European Nucleotide Archive under study ID PRJEB42056.

## SUPPLEMENTAL MATERIAL

Supplemental material is available online only.
**FIG S1**, TIF file, 2.1 MB.
**FIG S2**, TIF file, 1.7 MB.
**FIG S3**, TIF file, 2.6 MB.
**FIG S4**, TIF file, 1 MB.
**TABLE S1**, XLSX file, 0.01 MB.
**TABLE S2**, XLSX file, 0.01 MB.

## ACKNOWLEDGMENTS

We thank Qiyun Zhu for his assistance in performing the shotgun metagenomic taxonomy alignment.

C.M. was funded by NIDCR NRSA F31 Fellowship 1F31DE028478-01. K.J.C. is supported by NIH grant T32 DK007202.

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
