## [Reviewer comments · mSystems]

Evaluation of the effect of storage methods on fecal, saliva, and skin microbiome composition

Clarisse (Lisa) Marotz, Kellen Cavagnero, Se Jin Song, Daniel McDonald, Stephen Wandro, Greg Humphrey, MacKenzie Bryant, Gail Ackermann, Edgar Diaz, and Rob Knight

Corresponding Author(s): Rob Knight, UCSD School of Medicine

Review Timeline:

Submission Date:	December 18, 2020
Editorial Decision:	February 25, 2021
Revision Received:	March 15, 2021
Accepted:	March 23, 2021

Editor: Robert Beiko

Reviewer(s): The reviewers have opted to remain anonymous.

Transaction Report:

DOI: <https://doi.org/10.1128/mSystems.01329-20>

February 25, 2021

Prof. Rob Knight
UCSD School of Medicine
9500 Gilman Drive
MC 0602
La Jolla, CA 92093

Re: mSystems01329-20 (**Evaluation of the effect of storage methods on fecal, saliva, and skin microbiome composition**)

Dear Prof. Rob Knight:

I have two referee reviews in hand, and although they are generally favourable towards the work they raise a couple of very good points. Please revise in accordance with their comments, particularly around the use of weighted UniFrac and the greater emphasis on your results vs. the similar results of the Song paper.

Below you will find the comments of the reviewers.

To submit your modified manuscript, log onto the eJP submission site at <https://msystems.msubmit.net/cgi-bin/main.plex>. If you cannot remember your password, click the "Can't remember your password?" link and follow the instructions on the screen. Go to Author Tasks and click the appropriate manuscript title to begin the resubmission process. The information that you entered when you first submitted the paper will be displayed. Please update the information as necessary. Provide (1) point-by-point responses to the issues raised by the reviewers as file type "Response to Reviewers," not in your cover letter, and (2) a PDF file that indicates the changes from the original submission (by highlighting or underlining the changes) as file type "Marked Up Manuscript - For Review Only."

Due to the SARS-CoV-2 pandemic, our typical 60 day deadline for revisions will not be applied. I hope that you will be able to submit a revised manuscript soon, but want to reassure you that the journal will be flexible in terms of timing, particularly if experimental revisions are needed. When you are ready to resubmit, please know that our staff and Editors are working remotely and handling submissions without delay. If you do not wish to modify the manuscript and prefer to submit it to another journal, please notify me of your decision immediately so that the manuscript may be formally withdrawn from consideration by mSystems.

Sincerely,

Robert Beiko

Editor, mSystems

Journals Department
Reviewer comments:

Reviewer #1 (Comments for the Author):

This paper examines the effect of commonly-used preservatives on the microbial populations in stool, saliva and skin samples. In general, the data are convincing up to a certain point - 95% EtOH does seem fairly good at preserving many of the presented (although inappropriate for some) statistics/measurements, however other methods do seem to do as good a job, if not better (such as the OMNI method) in many cases and there is a general lack of discussion around spread of the data (which impart whether methods are reducing intrasample variability during storage) vs. departures from baselines. There is also a lack of statistical testing/support for some comments relating to differences between methods (see specific lines below). There are also some presentational/grammar problems that need to be addressed according to my comments below that seem to indicate a lack of careful proofreading of the manuscript and preparation of appropriate resolution figures.

However, a potential major problem in the manuscript is the exclusive reliance upon the unweighted UniFrac measure for all of the beta diversity analyses. Generally, I applaud the authors for using/prefer myself the use of UniFrac over a more simplified measure such as Bray-Curtis distances. However, as I'm sure the authors are aware, the unweighted version of UniFrac simply examines the common genera/species/ASVs between samples and completely ignores their relative abundances. There has also been recent discussion that the unweighted UniFrac measure is fundamentally flawed in many scenarios (Wong et al. 2016, PLoS ONE) of NGS sequencing analyses and that the weighted version (or two new versions of weightings) should consistently be used instead. In our research, we have opted to forgo the unweighted version and prefer the weighted UniFrac as the main display of beta diversity.

Regardless of whether you agree that the above potential flaw in unweighted UniFrac may be very significant or not, the exclusive use of only the unweighted version throughout the manuscript to argue that there are little changes in community composition due to preservatives is, at best, potentially misleading (depending on what the weighted results look like) and, at worse, hiding very significant differences. The effect of adding preservatives may, in fact, maintain all species originally present in the samples throughout storage until sequencing. However, the abundances of those

species may be completely altered from the original samples, yet your unweighted UniFrac analysis will show them to still be essentially identical. Most probably, different species will exhibit differing survivability in different preservatives and will therefore show different patterns of abundance skewness. Additionally, the exclusive use of the Shannon index for your alpha diversity measure may also gloss over these potential abundance skewness changes between communities as it is more influenced by shared richness vs., for example, the Simpson index which is more sensitive to changes in distributions.

You may be lucky, as hinted at by the limited changes (in some scenarios) of the 16S copy numbers as profiled by qPCR, that abundances are little affected. However, a drop of just 10% of the copy numbers can affect abundance compositions a fair amount in compositional data (as NGS data is) if they disproportionately affect certain dominant taxa. Weighted UniFrac is required to show whether the samples do, in fact, look quite similar to their original distributions or whether there have been significant alterations. We currently just do not have the analysis to make an informed conclusion in the manuscript's current state.

Specific comments

- 1) L.54-63: It is a bit usual that you have no referencing in this whole first paragraph, especially when listing specific examples such as field sampling from indigenous peoples.
- 2) L.116-118: Due to the spread of the data, it is highly unlikely that these differences are in fact significantly different from one another (for ex: OMNI vs. 95% EtOH). The poor resolution of the figure also makes it hard to conclude. If you are going to assert that actual increases have occurred here, you are going to need to test for difference from the baseline of 0 and inter-preservative difference. Additionally, just as important as "departure from baseline" is the overall spread of the data - methods such as OMNI and FTA seem to show smaller spread in the data than 95% EtOH meaning that they are better at controlling variability during storage.
- 3) L.165: I suspect here you ran a Principal Coordinates Analysis, which is the analysis type that belongs to the stated acronym of "PCoA", and not the stated Principal Components Analysis which should be abbreviated as "PCA" and is usually not used in the QIIME2/EMPeror plots.
- 4) L.240: You did not use more than one denoising tool.
- 5) L.248: Colloquial wording - change to "...the best choice was storing liquid saliva at a ratio of...".
- 6) L.281: You have a placeholder "cite" here instead of the expected reference.

Figure 1: Low resolution here is making the legends difficult to read. Panel B overall is too small and needs to be improved in resolution and/or size for better display.

Figure 2: The analysis in this figure is not from 16S OTUs, but the proportions of genera from metagenomics, so your axes labels are incorrect.

Figure 3: There is a "no title" extra shape on the figure which doesn't belong.

Figure 4: Verify your legend here - there are multiple cases of missing spaces between numbers, symbols and words.

General note in Supplemental Figures: All figures are of poor resolution when zoomed in and will need to be systematically increased in size to be correctly legible (esp. S2+S3).

Figure S5: Deblur is not an OTU caller, it is an ASV caller. You also need to replace the "OTU" labels on the axes for "ASV".

Reviewer #2 (Comments for the Author):

Review of Marotz et al manuscript entitled " Evaluation of the effect of storage methods on fecal, saliva, and skin microbiome composition 95% ethanol is a robust, cost-effective preservative"
General comments: The authors propose a universal method for the collection and storage of human microbiome studies. They expand on a previous study that had identified 95% ethanol as the preservative of choice for human microbiome collection for downstream DNA extraction and sequencing. The authors first re-analyzed the 16S amplicon sequence reads obtained from a previous study by Song et al 2016 using this time more recent pipelines that yields the more widely accepted amplicon sequence variants (ASVs) as opposed to operational taxonomic units (OTUs). In addition, they sequenced the stored extracted DNA to obtain shot-gun metagenomic reads. Both results confirmed the previous study by Song et al (2016), which recommended the use of 95% ethanol as a cost-effective preservative that can be used for citizen science and in remote areas that do not have direct access to a laboratory. The authors then proceed with a more novel aspect of their study designed to refine the efficacy of 95% ethanol preservation for fecal samples, saliva samples and skin samples, by adjusting the ratio of ethanol to sample. Finally, they made recommendations on the best conditions to preserve the samples with 95% ethanol, also indicating that while the recovery of the diversity is similar to the frozen "gold" standard for both the fecal and saliva samples, the skin samples were more difficult to recover from ethanol swabs compared to the frozen samples.

General comments: The authors identify the importance of the work as a methodological improvement for the collection of samples far away from the laboratory. The method is cost effective and allows storage at room temperature. Their work validates an earlier publication Song et al. 2016. The contribution of the first part of the manuscript, including Fig. 1 and Fig. 2 provides only an incremental advance over Song et al., even with the analysis of the metagenome reads, given that there is no new DNA extraction from the preserved samples but simply a re-analysis in the case of 16S reads and sequencing of previously extracted DNA for the metagenome. I recommend that Figure 1 and 2 be moved to the supplemental information section, and that the manuscript instead starts with the analysis of the new samples where the optimal preservation conditions are examined. Otherwise, the authors need to better justify the addition of Figure 1 and 2 to the main manuscript, beyond a simple confirmation of previous results. By moving the material that simply confirm the previous study to the supplemental information section, the authors would put in evidence the new findings and could also considerably reduce the text. It would also bring home the more important aspect of the study which is in the analysis of the different types of samples and the ratio of ethanol to sample to provide the best preservation.

Specific comments

Abstract:

Line 29-33: Reanalyzing extracted DNA is not really a breakthrough. DNA that is well isolated will keep its integrity for years when store correctly The novel aspect of the study should focus more on the preservation and analysis of the newly collected samples, i.e. the fecal, saliva and skin samples. In the abstract, the authors report on the optimal collection and storage protocol for fecal swab and saliva samples.

Figure 1 A: Move to supplemental information section. Large dot vs small dots are difficult to differentiate. Reference distance lines are also difficult to differentiate with just various shades of grey. It looks like the FTA provided the most consistent results when comparing fresh ambient and

preservation method.

Line 115-118: This statement should be associated with a figure It seems like an important point but there is no visualization available.

Line 129-132: To determine whether our diversity findings were consistent in terms of relative taxonomic abundance, we compared the relative abundance of taxa in each subject's fresh, preservative-treated samples to each subject's respective preservative-treated 8 week sample, for each preservative and temperature condition (Fig. 2A, Sup. Fig 4A)". This sentence is unclear and convoluted.

Line 156: The novel aspect of the manuscript start here with the optimization of the ratio of 95% ethanol preservative to sample. The previous section really detract from the new information.

Reviewer #1 (Comments for the Author):

This paper examines the effect of commonly-used preservatives on the microbial populations in stool, saliva and skin samples. In general, the data are convincing up to a certain point - 95% EtOH does seem fairly good at preserving many of the presented (although inappropriate for some) statistics/measurements, however other methods do seem to do as good a job, if not better (such as the OMNI method) in many cases and there is a general lack of discussion around spread of the data (which impart whether methods are reducing intrasample variability during storage) vs. departures from baselines. There is also a lack of statistical testing/support for some comments relating to differences between methods (see specific lines below). There are also some presentational/grammar problems that need to be addressed according to my comments below that seem to indicate a lack of careful proofreading of the manuscript and preparation of appropriate resolution figures.

We thank the reviewer for their thoughtful comments. In addition to addressing all presentational/grammar problems noted, we have included additional alpha and beta diversity calculations as suggested and feel that these changes have significantly improved our manuscript.

However, a potential major problem in the manuscript is the exclusive reliance upon the unweighted UniFrac measure for all of the beta diversity analyses. Generally, I applaud the authors for using/prefer myself the use of UniFrac over a more simplified measure such as Bray-Curtis distances. However, as I'm sure the authors are aware, the unweighted version of UniFrac simply examines the common genera/species/ASVs between samples and completely ignores their relative abundances. There has also been recent discussion that the unweighted UniFrac measure is fundamentally flawed in many scenarios (Wong et al. 2016, PLoS ONE) of NGS sequencing analyses and that the weighted version (or two new versions of weightings) should consistently be used instead. In our research, we have opted to forgo the unweighted version and prefer the weighted UniFrac as the main display of beta diversity.

Thank you for this comment. As suggested, we have updated all PCoA plots in this manuscript to show weighted UniFrac.

For thoroughness, we ran additional beta-diversity analyses which do not take into account phylogenetic relatedness are either commonly employed or with different distance metrics including Jaccard (non-phylogenetic presence/absence based), Bray Curtis (relative abundance weighted but non-phylogenetic), as well as two methods that take into account the compositional nature of microbiome sequencing data (Aitchison, RPCA). Remarkably, across all six metrics tested, the PERMANOVA test results clearly show that the host subject much more strongly drives beta-diversity separation compared to storage method. For both the shotgun sequencing re-analysis (Table S1) and the EtOH ratio experiments (Table S2) we have included tables showing the pseudo-F statistic and p-value from the PERMANOVA tests calculated from these six beta-diversity distance metrics.

Regardless of whether you agree that the above potential flaw in unweighted UniFrac may be very significant or not, the exclusive use of only the unweighted version throughout the manuscript to argue that there are little changes in community composition due to preservatives is, at best, potentially misleading (depending on what the weighted results look like) and, at worse, hiding very significant differences. The effect of adding preservatives may, in fact, maintain all species originally present in the samples throughout storage until sequencing. However, the abundances of those species may be completely altered from the original samples, yet your unweighted UniFrac analysis will show them to still be essentially identical. Most probably, different

species will exhibit differing survivability in different preservatives and will therefore show different patterns of abundance skewness. Additionally, the exclusive use of the Shannon index for your alpha diversity measure may also gloss over these potential abundance skewness changes between communities as it is more influenced by shared richness vs., for example, the Simpson index which is more sensitive to changes in distributions.

We have included a panel showing alpha-diversity metrics calculated with the Simpson index as recommended.

You may be lucky, as hinted at by the limited changes (in some scenarios) of the 16S copy numbers as profiled by qPCR, that abundances are little affected. However, a drop of just 10% of the copy numbers can affect abundance compositions a fair amount in compositional data (as NGS data is) if they disproportionately affect certain dominant taxa. Weighted UniFrac is required to show whether the samples do, in fact, look quite similar to their original distributions or whether there have been significant alterations. We currently just do not have the analysis to make an informed conclusion in the manuscript's current state.

We believe that the additional analyses performed in response to your comments has greatly strengthened our manuscript and our ability to determine that 95% EtOH is a reasonable preservation method for many microbiome study designs.

Specific comments

1) L.54-63: It is a bit usual that you have no referencing in this whole first paragraph, especially when listing specific examples such as field sampling from indigenous peoples.

We have included multiple references in this introduction paragraph, including a recent study that describes the generation of one of the largest curated microbial genome database to date and specifically identifies geographic regions that contain some of the highest amounts of microbial diversity but are under-represented (Almeida et al., 2021 *Nat Biotech*).

2) L.116-118: Due to the spread of the data, it is highly unlikely that these differences are in fact significantly different from one another (for ex: OMNI vs. 95% EtOH). The poor resolution of the figure also makes it hard to conclude. If you are going to assert that actual increases have occurred here, you are going to need to test for difference from the baseline of 0 and inter-preservative difference. Additionally, just as important as "departure from baseline" is the overall spread of the data - methods such as OMNI and FTA seem to show smaller spread in the data than 95% EtOH meaning that they are better at controlling variability during storage.

Thank you for highlighting this. As recommended we have additionally performed alpha-diversity calculations with the Simpson index, and this is now included in Supplemental Figure 3 alongside the Shannon diversity calculations. We have updated the text to clarify that no preservative and 70% EtOH preservation had the largest changes in alpha-diversity, while all other preservatives performed similarly well:

“As with composition, differences in Shannon and Simpson alpha diversity between non-stored and stored samples were largest when samples were unfixed or stored in 70% EtOH, and smallest when stored in 95% EtOH, OMNI-gene GUT, FTA, and RNAlater (Sup. Fig. 2)”

3) L.165: I suspect here you ran a Principal Coordinates Analysis, which is the analysis type that belongs to the stated acronym of "PCoA", and not the stated Principal Components Analysis which should be abbreviated as "PCA" and is usually not used in the QIIME2/EMPeror plots.

4) L.240: You did not use more than one denoising tool.

5) L.248: Colloquial wording - change to "...the best choice was storing liquid saliva at a ratio of...".

6) L.281: You have a placeholder "cite" here instead of the expected reference.

Thank you for catching these errors, we have updated the manuscript accordingly.

Figure 1: Low resolution here is making the legends difficult to read. Panel B overall is too small and needs to be improved in resolution and/or size for better display.

Thank you for noting this; we have increased the resolution and the size of the axes/legends.

Figure 2: The analysis in this figure is not from 16S OTUs, but the proportions of genera from metagenomics, so your axes labels are incorrect.

We have updated the axes labels to 'gOTU abundance' in accordance with the shotgun sequencing analysis performed here. This terminology (genome OTU, 'gOTU') was coined because of the marker gene approach used to annotate shotgun sequencing datasets (<https://github.com/qiyunzhu/woltka>). We have also added this detail into the methods section for clarification.

Figure 3: There is a "no title" extra shape on the figure which doesn't belong.

Thank you for catching this - we have updated figures 3 and increased the resolution.

Figure 4: Verify your legend here - there are multiple cases of missing spaces between numbers, symbols and words.

The legend has been updated to consistently include a space between numbers and their units.

General note in Supplemental Figures: All figures are of poor resolution when zoomed in and will need to be systematically increased in size to be correctly legible (esp. S2+S3).

We have increased the resolution of all figures.

Figure S5: Deblur is not an OTU caller, it is an ASV caller. You also need to replace the "OTU" labels on the axes for "ASV".

We have updated this legend to read "ASV" rather than OTU.

Reviewer #2 (Comments for the Author):

Review of Marotz et al manuscript entitled " Evaluation of the effect of storage methods on fecal, saliva, and skin microbiome composition 95% ethanol is a robust, cost-effective preservative"

General comments: The authors propose a universal method for the collection and storage of human microbiome studies. They expand on a previous study that had identified 95% ethanol as the preservative of choice for human microbiome collection for downstream DNA extraction and sequencing. The authors first re-analyzed the 16S amplicon sequence reads obtained from a previous study by Song et al 2016 using this time more recent pipelines that yields the more widely accepted amplicon sequence variants (ASVs) as opposed to operational taxonomic units (OTUs). In addition, they sequenced the stored extracted DNA to obtain shot-gun metagenomic reads. Both results confirmed the previous study by Song et al (2016), which recommended the use of 95% ethanol as a cost-effective preservative that can be used for citizen science and in remote areas that do not have direct access to a laboratory. The authors then proceed with a more novel aspect of their study designed to refine the efficacy of 95% ethanol preservation for fecal samples, saliva samples and skin samples, by adjusting the ratio of ethanol to sample. Finally, they made recommendations on the best conditions to preserve the samples with 95% ethanol, also indicating that while the recovery of the diversity is similar to the frozen "gold" standard for both the fecal and saliva samples, the skin samples were more difficult to recover from ethanol swabs compared to the frozen samples.

General comments: The authors identify the importance of the work as a methodological improvement for the collection of samples far away from the laboratory. The method is cost effective and allows storage at room temperature. Their work validates an earlier publication Song et al. 2016. The contribution of the first part of the manuscript, including Fig. 1 and Fig. 2 provides only an incremental advance over Song et al., even with the analysis of the metagenome reads, given that there is no new DNA extraction from the preserved samples but simply a re-analysis in the case of 16S reads and sequencing of previously extracted DNA for the metagenome. I recommend that Figure 1 and 2 be moved to the supplemental information section, and that the manuscript instead starts with the analysis of the new samples where the optimal preservation conditions are examined. Otherwise, the authors need to better justify the addition of Figure 1 and 2 to the main manuscript, beyond a simple confirmation of previous results. By moving the material that simply confirm the previous study to the supplemental information section, the authors would put in evidence the new findings and could also considerably reduce the text. It would also bring home the more important aspect of the study which is in the analysis of the different types of samples and the ratio of ethanol to sample to provide the best preservation.

Thank you for this comment. We agree with you that we should reduce the amount of text and figures devoted to the re-sequencing and analysis of the Song et al., samples. However, we feel it is important to show that the shotgun sequencing results recapitulate the 16S results given the technical differences in library preparation. Therefore we have combined figures 1 and 2 and reduced the text describing these results so that more time can be focused on the EtOH ratio experimental data.

Specific comments

Abstract:

Line 29-33: Reanalyzing extracted DNA is not really a breakthrough. DNA that is well isolated will keep its integrity for years when store correctly The novel aspect of the study should focus more on the preservation and analysis of the newly collected samples, i.e. the fecal, saliva and skin samples. In the abstract, the authors report on the optimal collection and storage protocol for fecal swab and saliva samples.

We have compressed figures 1 and 2 into a single figure and reduced the text describing this re-analysis.

Figure 1 A: Move to supplemental information section. Large dot vs small dots are difficult to differentiate. Reference distance lines are also difficult to differentiate with just various shades of grey. It looks like the FTA provided the most consistent results when comparing fresh ambient and preservation method.

Thank you for these comments. Instead of using large versus small dots in figure 1A, we have labeled the fresh samples as diamonds which are easier to distinguish. We have also increased the size of panel 1B so that the lines are easier to differentiate. FTA, 95% EtOH, OMNI, and RNA later all performed well to maintain microbial composition, but of these preservatives 95% EtOH is the most practical and affordable. We have highlighted this point at the end of the first section of the results.

Line 115-118: This statement should be associated with a figure It seems like an important point but there is no visualization available.

This statement is now supported by Sup. Fig. S2. In an effort to minimize the text we have updated this section on alpha-diversity analysis to read:

“As with composition, differences in Shannon and Simpson alpha diversity between non-stored and stored samples were largest when samples were unfixed or stored in 70% EtOH, and smallest when stored in 95% EtOH, OMNI-gene GUT, FTA, and RNA later (Sup. Fig. 2)

Line 129-132: To determine whether our diversity findings were consistent in terms of relative taxonomic abundance, we compared the relative abundance of taxa in each subject's fresh, preservative-treated samples to each subject's respective preservative-treated 8 week sample, for each preservative and temperature condition (Fig. 2A, Sup. Fig 4A)". This sentence is unclear and convoluted.

Thank you for pointing this out, we have updated this sentence to read:

"To identify potential changes in the relative abundance of specific taxonomic clades, we compared the relative abundance of each genus in each subject's sample processed fresh versus 8 weeks out for each preservative (Fig. 1C, Sup. Fig 1C)"

Line 156: The novel aspect of the manuscript start here with the optimization of the ratio of 95% ethanol preservative to sample. The previous section really detract from the new information.

We have shortened this first section and compressed the first two figures to more clearly highlight the results from the ratio of 95% EtOH to sample experiments.

March 23, 2021

Prof. Rob Knight
UCSD School of Medicine
9500 Gilman Drive
MC 0602
La Jolla, CA 92093

Re: mSystems01329-20R1 (**Evaluation of the effect of storage methods on fecal, saliva, and skin microbiome composition**)

Dear Prof. Rob Knight:

Your manuscript has been accepted, and I am forwarding it to the ASM Journals Department for publication. For your reference, ASM Journals' address is given below. Before it can be scheduled for publication, your manuscript will be checked by the mSystems senior production editor, Ellie Ghatineh, to make sure that all elements meet the technical requirements for publication. She will contact you if anything needs to be revised before copyediting and production can begin. Otherwise, you will be notified when your proofs are ready to be viewed.

- Minimum resolution of 1280 x 720
- .mov or .mp4. video format
- Provide video in the highest quality possible, but do not exceed 1080p
- Provide a still/profile picture that is 640 (w) x 720 (h) max

We recognize that the video files can become quite large, and so to avoid quality loss ASM suggests sending the video file via <https://www.wetransfer.com/>. When you have a final version of the video and the still ready to share, please send it to Ellie Ghatineh at eghatineh@asmusa.org.

Sincerely,

Robert Beiko
Editor, mSystems

Journals Department
Fig. S2: Accept
Fig. S4: Accept
Table S2: Accept
Table S1: Accept
Fig. S3: Accept
Fig. S1: Accept